# Shaped by the COVID-19 pandemic: Psychological responses from a subjective perspective–A longitudinal mixed-methods study across five European countries

**Irina Zrnić Novaković**[1,2]*, **Dean Ajduković**[3], **Helena Bakić**[3], **Camila Borges**[4], **Margarida Figueiredo-Braga**[4,5], **Annett Lotzin**[6,7], **Xenia Anastassiou-Hadjicharalambous**[8], **Chrysanthi Lioupi**[8], **Jana Darejan Javakhishvili**[9], **Lela Tsiskarishvili**[10], **Brigitte Lueger-Schuster**[1]

1 Department of Clinical and Health Psychology, Faculty of Psychology, University of Vienna, Vienna, Austria, 2 Vienna Doctoral School in Cognition, Behaviour and Neuroscience, University of Vienna, Vienna, Austria, 3 Department of Psychology, Faculty of Humanities and Social Sciences, University of Zagreb, Zagreb, Croatia, 4 Trauma Observatory, Centre for Social Studies (CES) of the University of Coimbra, Coimbra, Portugal, 5 Department of Clinical Neurosciences and Mental Health, Faculty of Medicine, University of Porto, Porto, Portugal, 6 Department of Psychiatry and Psychotherapy, University Medical Center Hamburg-Eppendorf, Hamburg, Germany, 7 Department of Psychology, MSH Medical School Hamburg, Hamburg, Germany, 8 Department of Social Sciences, School of Humanities, Social Sciences and Law, University of Nicosia, Nicosia, Cyprus, 9 Institute of Addiction Studies, School of Arts and Sciences, Ilia State University, Tbilisi, Georgia, 10 School of Arts and Sciences, Ilia State University, Tbilisi, Georgia

* irina.zrnic@univie.ac.at

**Data Availability Statement:** Both the minimal data set underlying the quantitative results and the

## Abstract

### Background

Contextual factors are essential for understanding long-term adjustment to the COVID-19 pandemic. Therefore, the present study investigated changes in mental health outcomes and subjective pandemic-related experiences over time and across countries. The main objective was to explore how psychological responses vary in relation to individual and environmental factors.

### Methods

The sample consisted of $N = 1070$ participants from the general population of Austria, Croatia, Georgia, Greece, and Portugal. We applied a longitudinal mixed-methods approach, with baseline assessment in summer and autumn 2020 (T1) and follow-up assessment 12 months later (T2). Qualitative content analysis by Mayring was used to analyse open-ended questions about stressful events, positive and negative aspects of the pandemic, and recommendations on how to cope. Mental health outcomes were assessed with the Adjustment Disorder–New Module 8 (ADNM-8), the Primary Care PTSD Screen for DSM-5 (PC-PTSD-5), the Patient Health Questionnaire-2 (PHQ-2), and the 5-item World Health Organization Well-Being Index (WHO-5). The analyses were performed with SPSS Statistics Version 26 and MAXQDA 2022.

relevant excerpts of the qualitative analysis can be found in the OSF repository under this link: https://osf.io/p9jx2/.

**Funding:** The authors received no specific funding for conducting this worky. Open access funding provided by the University of Vienna.

**Competing interests:** The authors have declared that no competing interests exist.

## Results

The mental health outcomes significantly differed over time and across countries, with e.g. Greek participants showing decrease in adjustment disorder symptoms ($p = .007$) between T1 and T2. Compared with other countries, we found better mental health outcomes in the Austrian and the Croatian sample at both timepoints ($p < .05$). Regarding qualitative data, some themes were equally represented at both timepoints (e.g. *Restrictions and changes in daily life*), while others were more prominent at T1 (e.g. *Work and finances*) or T2 (e.g. *Vaccination issues*).

## Conclusions

Our findings indicate that people's reactions to the pandemic are largely shaped by the shifting context of the pandemic, country-specific factors, and individual characteristics and circumstances. Resource-oriented interventions focusing on psychological flexibility might promote resilience and mental health amidst the COVID-19 pandemic and other global crises.

## Introduction

The COVID-19 pandemic has hugely challenged humans' ability to adjust. Whereas the mental health consequences of the pandemic have been studied intensively, little is known about how people perceive different pandemic phases and which factors shape their subjective experiences. Therefore, the present study investigates how personal perceptions change over time, and which factors underlie these changes, across five European countries.

### Longitudinal changes in mental health during the COVID-19 pandemic

The coronavirus disease 2019 (COVID-19) has been impacting our lives since March 2020, when the World Health Organization (WHO) declared it a global pandemic [1]. There is now a considerable amount of literature on the detrimental impacts of the pandemic on mental health [2], with numerous studies [3, 4] reporting high prevalences of anxiety, depression, and posttraumatic stress disorder (PTSD). Moreover, though not yet extensively investigated, symptoms of adjustment disorder (AD) appear to be on the rise [5, 6].

With regard to mental health trajectories over the course of the pandemic, research has yielded mixed findings. While several studies found that mental health has deteriorated over the duration of the pandemic [7, 8], there is also evidence to suggest resilience and improvements in mental health [9]. For instance, a study examining women living in the United States reported a decrease in stress and loneliness over 13 months of the pandemic [10].

Notably, different trajectories have been demonstrated for different mental disorders. For depression, an Italian study covering 10 months of the pandemic found a quadratic growth, with an initial rise followed by a decrease with the easing of the first lockdown and a subsequent increase during the second lockdown [11]. When compared with pre-pandemic data, a study in Germany reported an increase in anxiety and depressive symptoms during the first wave of COVID-19 (spring 2020) but a decrease during the second wave (January-February 2021; [12]). This stabilisation of anxiety and depressive symptoms was also reported in the USA [13]. By contrast, a recent Swiss study found that the prevalence of anxiety and depression almost doubled between August 2020 and May 2021 [14].

With regard to PTSD, a downward trend was observed in the first months of the pandemic in Spain [15] and in the second year of the pandemic in Germany, Israel, Poland, and Slovenia [16]. An Italian study [17] also described a decrease in posttraumatic symptoms between March and May 2020. Notably, the same study observed a significant increase in November 2020, when the second wave of infection started, and new restrictions were introduced in Italy.

In terms of AD, an Austrian study covering two years of the pandemic detected higher mean scores and prevalence in wintertime, characterised by stricter regulations, higher incidence and higher death rates compared with summertime [18]. For *preoccupation* and *failure to adapt* as core symptoms of AD, a significant linear decrease was observed in a Canadian study conducted over the spring of 2020 [19].

Changes in well-being have also been reported. For instance, studies conducted over the first year of pandemic reported a decline in well-being over time [20, 21]. When comparing well-being in spring 2020 and spring 2021, different patterns were found in different countries, with slightly more participants showing improved rather than deteriorated well-being [22].

In sum, fluctuations in mental health seem to depend on the investigated outcomes, but also on the characteristics of the sample. Certain groups have experienced more difficulties in adapting to the pandemic, reporting more mental health problems as the pandemic has progressed. For example, younger people have shown poorer mental health over the course of the pandemic [23, 24]. The restrictions of social and leisure activities as well as disruptions to education and the job market have severely impacted this age group, resulting in lower life satisfaction, more loneliness, and greater anxiety and depressive symptoms [12].

Besides younger people, women and individuals with lower income are particularly burdened during COVID-19. The pandemic seems to have exacerbated gender inequalities, e.g. regarding the division of housework and childcare, putting women at higher risk of developing persisting mental health problems [25, 26]. Furthermore, the pandemic has impacted the global economy, leading to numerous job and income losses [27]. Not only at the beginning of the pandemic [28], but also in later phases, financial difficulties and female gender have been associated with a greater mental health burden [8, 29].

Taken together, psychological reactions to the pandemic over time are impacted by different factors. The fluctuations in mental health seem to correspond to the containment measures in place at the time of assessment [17], especially social restrictions [30], but might also be related to seasonal effects [18]. Moreover, international studies have shown that mental health outcomes are country-dependent [31]. In addition to environmental factors, personal characteristics (e.g. trait boredom) might also be responsible for changes in mental health [19].

## Mechanisms underlying changes in mental health

According to the WHO framework for the conceptual determinants of health [32], an unfavourable combination of environmental factors, and personal and sociodemographic characteristics might increase an individual's vulnerability to develop mental health problems (for a framework of health in the context of COVID-19, see [33]). Notably, individuals who have suffered pandemic-related resource losses (e.g. people who have lost their income, young people who have lost social connections, women who have lost family stability due to a work-family imbalance) are particularly vulnerable to mental health problems and loss spirals, as proposed by the *conservation of resources (COR) theory* [34, 35].

The mental health impact of COVID-19 is also associated with how people perceive the pandemic and cope with pandemic-related stressors [36, 37]. Perceiving more stress in later waves than in the initial wave of the pandemic was associated with lower well-being and life satisfaction and higher anxiety [38]. On the other hand, acknowledging and accepting distress,

and engaging in actions that are in line with one's own values despite the distress, were linked to better mental health outcomes [11]. This type of psychological response falls under the concept of *psychological flexibility*, i.e. humans' ability to adjust their behaviour in response to changing situational demands [39, 40].

## Aims and research questions of the present study

Overall, the interplay of subjective perceptions, coping behaviours, and individual and environmental factors might be essential for explaining psychological adjustment to the pandemic [41]. To better understand the mechanisms underlying varying mental health trajectories, the present study focused on how the COVID-19 pandemic has impacted different target groups in different phases of the pandemic. The overarching aim was to shed light on pandemic-related experiences over time and across countries, and to examine whether these experiences vary as a function of different personal and environmental factors. The specific aims were as follows:

1. Explore differences in mental health outcomes (symptoms of AD, PTSD and depression) across five European countries over 12 months of the COVID-19 pandemic.

2. Examine whether and how pandemic-related experiences have changed over time, specifically:

○ How has the experience of stressful events during the pandemic changed over time?

○ How have positive and negative aspects of the pandemic changed over time?

○ How have population-informed recommendations for dealing with the pandemic changed over time?

3. Examine whether pandemic-related experiences have differed between different groups (e.g. different countries, different age groups), specifically:

○ Have pandemic-related experiences differed depending on: age, gender, health-related characteristics (health status, being infected with COVID-19, being at risk of a severe course of COVID-19, history of mental health problems), social factors (face-to-face and virtual contact with loved ones, time spent at home, living situation), financial situation (pandemic-related income loss, financial support) and country of residence (Austria, Croatia, Georgia, Greece, Portugal)?

First, we hypothesised that mental health outcomes would follow different trajectories in different countries, depending on different contextual factors (e.g. socioeconomic situation in the respective country, containment measures in place). We further assumed that pandemic-related experiences would change over time, echoing the changing circumstances of the pandemic (e.g. differing stringency of containment measures). Finally, in view of the country-specific factors (e.g. availability of financial support from the government) and individual characteristics (e.g. health status), we hypothesised that pandemic-related experiences would differ between groups. For instance, we assumed that men and women would report different stressful events experienced during the pandemic.

## Methods

The present study applied a convergent design in a multistage mixed-methods framework [42]. In particular, we collected and analysed quantitative and qualitative data during a similar timeframe, at two stages of data assessment.

### Research context

The data used in this paper stem from the ADJUST study–a longitudinal pan-European study investigating stressors and risk and protective factors for AD amid COVID-19 [5, 33]. Before its start, the ADJUST study was registered in a study registry (https://osf.io/8xhyg). For the secondary analysis described here, we applied a mixed-methods design to compare the data collected in Austria, Croatia, Georgia, Greece, and Portugal in the first (15/06/2020–14/12/2020) and third (21/06/2021–14/12/2021) wave of the ADJUST study. The third wave will hereinafter be referred to as T2, as it represents the second measurement point of the present analysis. S1 Appendix provides more details on the COVID-19 situation in each country during the data assessment.

Both assessments were conducted as online surveys. After receiving information about the study aims, data management, confidentiality, and right to withdraw, all participants provided written informed consent. Ethical approval was obtained in all participating countries:

Austria: Ethics Committee of the University of Vienna, Reference Number: 00554

Croatia: Ethics Committee of the Department of Psychology, Faculty of Humanities and Social Sciences, University of Zagreb: 21/05/2020

Georgia: Ilia State University, Faculty of Arts and Science, Research Ethics Committee: 12/06/2020

Greece: Social Sciences Ethics Review Board (SSERB), University of Nicosia: SSERB 00109

Portugal: Ethics Committee of the Medical Faculty, University of Porto and Centro Hospitalar São João, Porto, Portugal: CE 201–20

### Participants and procedure

For T1, participants were recruited from the general population of Austria, Croatia, Georgia, Greece, and Portugal. We promoted the study via social media and approached several professional and hobby associations, psychosocial services, and large companies, asking them to share the study information including the survey link among their employees (see [5] for more information on the recruitment). Participants who completed the baseline assessment and agreed to be re-contacted for further assessments received two invitations for follow-up at intervals of approximately six months. This secondary analysis encompassed the baseline assessment (T1) and the 12-month follow-up assessment (T2).

Eligibility criteria for the present study included: (a) age ≥18 years, (b) living in one of the participating countries, (c) ability to read and write in the respective language, (d) willingness to participate, and (e) having responded to all relevant measures (i.e. closed- and open-ended questions) at the baseline and follow-up assessment.

The final sample consisted of $N$ = 1,070 participants. At baseline, the mean age was 42.9 years ($SD$ = 13.5) and the majority of the sample was female (74.4%, $n$ = 796). Sociodemographic characteristics are detailed in Table 1.

### Measures

The measures included several well-known questionnaires, a series of self-constructed items, and four open-ended questions. In the following, only the measures necessary for the present secondary analysis are described. An exhaustive overview of variables collected within the ADJUST study is provided elsewhere [5, 33].

**Quantitative data assessment.** To provide context for the longitudinal subgroup analysis, personal and environmental factors were assessed. In addition to sociodemographic characteristics (Table 1) and country of residence (Austria, Croatia, Georgia, Greece, and Portugal),

**Table 1. Overview of participants' sociodemographic characteristics at baseline (T1).**

| | Austria *n* = 333 | Croatia *n* = 414 | Georgia *n* = 113 | Greece *n* = 122 | Portugal *n* = 88 |
|---|---|---|---|---|---|
| | *M* (*SD*) | *M* (*SD*) | *M* (*SD*) | *M* (*SD*) | *M* (*SD*) |
| Age | 46.7 (14.8) | 42.8 (12.1) | 36.1 (13.3) | 37.5 (11.0) | 45.7 (11.6) |
| | *n* (%) | *n* (%) | *n* (%) | *n* (%) | *n* (%) |
| Gender | | | | | |
| Male | 106 (31.8) | 86 (20.8) | 19 (16.8) | 39 (32.0) | 22 (25.0) |
| Female | 225 (67.6) | 328 (79.2) | 94 (83.2) | 83 (68.0) | 66 (75.0) |
| Other | 2 (0.6) | 0 (0) | 0 (0) | 0 (0) | 0 (0) |
| Living situation [a] | | | | | |
| Living alone | 109 (32.7) | 94 (22.7) | 20 (17.7) | 33 (27.0) | 28 (31.8) |
| Living with parents | 10 (3.0) | 75 (18.1) | 59 (52.2) | 24 (19.7) | 8 (9.1) |
| Living with partner | 106 (31.8) | 89 (21.5) | 11 (9.7) | 36 (29.5) | 20 (22.7) |
| Living with partner and children | 78 (23.4) | 148 (35.7) | 30 (26.5) | 27 (22.1) | 24 (27.3) |
| Living with children | 19 (5.7) | 32 (7.7) | 3 (2.7) | 4 (3.3) | 2 (2.3) |
| Living with colleagues/ fellow students | 5 (1.5) | 4 (1.0) | 5 (4.4) | 1 (0.8) | 3 (3.4) |
| Other | 19 (5.7) | 27 (6.5) | 0 (0) | 13 (10.7) | 9 (10.2) |
| Relationship status | | | | | |
| Single | 78 (23.4) | 90 (21.7) | 47 (41.6) | 27 (22.1) | 21 (23.9) |
| Short-term relationship(s) | 11 (3.3) | 4 (1.0) | 3 (2.7) | 2 (1.6) | 1 (1.1) |
| Stable relationship -living separately | 45 (13.5) | 46 (11.1) | 15 (13.3) | 19 (15.6) | 4 (4.5) |
| Stable relationship—living together | 199 (59.8) | 274 (66.2) | 48 (42.5) | 74 (60.7) | 62 (70.5) |
| Children | | | | | |
| Yes | 190 (57.1) | 256 (61.8) | 44 (38.9) | 50 (41.0) | 48 (54.5) |
| No | 143 (42.9) | 158 (38.2) | 69 (61.1) | 72 (59.0) | 40 (45.5) |

[a] Percentages sum to more than 100 as multiple responses were possible.

health-related characteristics, social factors and financial situation were assessed using self-constructed items, as detailed in the S2 Table in S1 Appendix.

To explore the differences in mental health outcomes (Aim 1), we used established self-report questionnaires at both time points:

The Adjustment Disorder New Module—8 (ADNM-8; [43]) was used to measure symptoms of AD. It consists of eight items rated on a 4-point Likert scale (1 = *never* to 4 = *often*) and has shown good reliability and convergent and factorial validity [44]. In the present study, the ADNM-8 showed very high internal consistency, with Cronbach's α = .91 at baseline and .92 at follow-up.

The Primary Care PTSD Screen for DSM-5 (PC-PTSD-5) is a widely used screening tool for PTSD [45, 46]. It contains five items assessing the presence of posttraumatic stress symptoms according to the DSM-5 with a dichotomous response format. In the present study, we included in the analysis only the PC-PTSD-5 scores of the participants who reported a traumatic event during the pandemic. Cronbach's α of the PC-PTSD-5 was acceptable at both time points of our study (.74 and .76).

The Patient Health Questionnaire-2 (PHQ-2;) is a two-item depression screener with good psychometric properties [47, 48]. It assesses the most common symptoms of depression on a 4-point Likert scale ranging from 0 = n*ot at all* to 3 = *nearly every day*. In the present study, the PHQ-2 showed high internal consistency (Cronbach's α = .80 at T1 and .86 at T2).

The 5-item World Health Organization Well-Being Index (WHO-5; [49]) measures well-being over the last two weeks using a 5-point Likert scale (0 = *at no time* to 5 = *all of the time*).

It has shown high validity and applicability across various contexts [50]. Internal consistency of the WHO-5 in our study was very high, with Cronbach's α = .90 at baseline and .91 at follow-up.

**Qualitative data assessment.** To assess subjective pandemic-related experiences in different pandemic stages (Aim 2), the following open-ended questions were asked at T1 and T2:

1. What has been the most stressful event(s) during the coronavirus pandemic?

2. Overall, what do you find most negative about the coronavirus pandemic?

3. Overall, what do you find most positive about the coronavirus pandemic?

4. What recommendations would you give to other people on how to deal with the current situation?

## Data analysis

For the data analysis, we chose a longitudinal mixed-methods approach, enabling us to investigate mental health trajectories and evolving pandemic-related experiences from multiple perspectives. This approach is widely accepted and well suited to analyse experiences of change and stability shaped by contextual factors [51, 52], as is the case in our study. A complete case analysis was performed given that both quantitative and qualitative data had to be analysed longitudinally and it is impossible to circumvent missing qualitative data, e.g. using imputation. Thus, all qualitative data could be integrated with the corresponding quantitative data to yield triangulated findings, increasing the validity of the study and offering deeper insights into how and why mental health changes in relation to personal and environmental factors.

**Quantitative data analysis.** First, all independent variables and mental health outcomes were analysed descriptively to provide a differentiated picture of the sample, in view of the importance of contextual information in longitudinal mixed-methods research. To analyse mean or median differences in independent variables over time, we performed *t*-tests, McNemar tests [53] with continuity correction [54], or Wilcoxon signed-rank tests, as appropriate. Normal Q-Q Plots were inspected to check for normality as the sample size was greater than 50.

For Aim 1, the purpose of the analysis was to provide a general understanding of the development of mental health symptoms during the pandemic and of possible differences between the participating countries. Differences in AD, PTSD, depression, and well-being scores between T1 and T2 were analysed for each country using paired-samples *t*-tests or Wilcoxon signed-rank tests, and the estimated prevalence rates were compared using a McNemar test. In addition, one-way analysis of variance (ANOVA) with Tukey post hoc test was used for each time point to test the differences in mental health outcomes between the participating countries at T1 and T2, respectively. Welch's ANOVAs and Games-Howell post hoc tests were applied if the assumption of homogeneity of variances was violated, as assessed with the Levene's test. Depending on the analysis, different measures of effect size (Cohen's *d*, Odds ratio [OR], Rosenthal's *r*, Omega squared [$\omega^2$]) were reported [55]. All analyses were conducted using SPSS Statistics Version 26 [56].

**Qualitative data analysis.** For Aims 2 and 3, qualitative content analysis (QCA) was used to analyse the four open-ended questions [57, 58]. This method follows strict procedural rules and is recommended for large samples. QCA also includes the use of frequencies (i.e. semi-quantification), which can highlight the relevance of the identified themes and categories and are essential in a mixed-methods analysis [59, 60]. QCA thus allows for a juxtaposition of answers provided by different target groups (e.g. participants from different countries) at T1

and T2, which is necessary to identify differences and recognise patterns. All analyses involving qualitative data were performed using MAXQDA 2022 [61].

In a first step, we analysed participants' answers collected at T1. The Austrian study team, which initiated the study, developed a preliminary coding scheme based on randomly selected answers of $n = 180$ Austrian participants (i.e. using an inductive approach). Next, the remaining countries applied this coding scheme to code their data (i.e. using a deductive approach in their respective languages). The coding system was finalised in an iterative process of category revision, and final categories and themes (i.e. main categories) were developed. Further procedural details can be found in another paper of the research group [62] and in the respective OSF storage (https://osf.io/6njdu/).

The data collected at T2 were analysed in a deductive manner. All countries applied the existing T1 coding scheme to their data collected at T2 to ensure consistency in the analysis and to enable the data to be compared over time. However, additional categories could be added after discussion in the research group if segments were identified which did not correspond to any of the existing categories. After the coding was finished, the categories were grouped into themes, in accordance with the procedure at T1. For the purpose of reporting the results, selected quotes were translated into English only after the coding process was finalised.

Inter- and intrarater agreement were calculated based on the answers of 10% randomly selected participants, resulting in a high level of agreement in all countries at both timepoints (for details, see S1 Appendix).

**Mixed-methods analysis.** In a final step (Aim 3), the T1 and T2 data were merged into one common MAXQDA project. Relevant quantitative data (i.e. health-related characteristics, social factors, financial situation; detailed in S1 Appendix) were then imported into the common project. The themes identified at T1 and T2 were quantitatively and qualitatively compared using the respective MAXQDA functions. For instance, *crosstabs* were used to compare frequencies of the identified themes at T1 and T2. In addition, themes were examined with regard to statistical characteristics using *joint displays* (e.g. to determine the mean age of participants reporting emotional distress as a negative aspect of the pandemic).

Throughout the research process, we followed the existing guidelines for cross-country and longitudinal mixed-methods research [63–66]. We integrated the data (1) at the *study design level* by applying a convergent, multistage design; (2) at the *methods level* by merging the qualitative and quantitative data for analysis and comparison; and (3) at the *interpretation level* through narratives, data transformation (e.g. semi-quantification of qualitative data) and joint displays (see Material and methods) to enhance the quality and value of our research [42]. Regular meetings were held to reflect on the coding process and jointly discuss the coding scheme, definitions of the categories, and development of themes.

## Results

A total of $N = 5,405$ participants from Austria (AUT), Croatia (CRO), Georgia (GEO), Greece (GR), and Portugal (PT) responded to all relevant closed- and open-ended questions at baseline. We excluded participants who were either unwilling to participate in the follow-up or who did not respond to some of the relevant measures at follow-up. The final sample consisted of $N = 1,070$ participants.

### Sample characteristics

At baseline, the majority of participants reported being in a relationship (75.4%; $n = 807$) and having children (55%; $n = 588$). Across the countries, around a quarter of participants indicated living alone (26.5%, $n = 284$), with the highest percentage in Austria (32.7%, $n = 109$)

and the lowest in Georgia (17.7%, $n = 20$). Sociodemographic characteristics for each country are shown in Table 1.

**Health-related characteristics.** Participants described their current health as better at baseline ($M = 1.92$, $SD = 0.83$) than at follow-up ($M = 2.04$, $SD = 0.86$), $t(1069) = -4.83$, $p < .001$, $d = 0.148$. Besides, more participants described their health as *very good* at baseline (35.2%, $n = 377$) than at follow-up (28.9%, $n = 309$)

With regard to the diagnosis of a mental disorder, there was no significant median difference between T1 and T2 ($z = -1.07$, $p = .29$, $r = -.03$). At T1, 83.6% of participants ($n = 894$) reported not having a diagnosed mental disorder, whereas this was true for 82.5% of participants at T2 ($n = 883$).

The proportion of participants who had been infected with COVID-19 significantly increased over time ($\chi^2(1) = 166.05$, $p < .001$, OR = 171), as shown by a McNemar test. While only six participants (0.6%) reported a COVID-19 infection at T1, $n = 176$ participants did so at T2 (16.4%).

Participants also evaluated their risk of severe symptoms of COVID-19 differently between T1 and T2 ($\chi^2(1) = 18.45$, $p < .001$, OR = 0.49). At baseline, 19.9% ($n = 213$) reported being at risk of severe symptoms of COVID-19, as opposed to 14.7% ($n = 157$) at follow-up.

Participants' health-related characteristics per time point, divided by country, are depicted in S4 Table in S1 Appendix.

**Social factors.** At follow-up, $n = 312$ (29.2%) reported not spending more time at home, compared to $n = 117$ (10.9%) at baseline. This change was mainly a consequence of 247 people (23.1%) spending more time at home at baseline but not at follow-up. The difference was statistically significant, as demonstrated by a McNemar test, $\chi^2(1) = 125.87$, $p < .001$, OR = 4.75.

The participants tended to have more face-to-face contact with loved ones at follow-up than at baseline, with the median being significantly higher at follow-up according to a Wilcoxon test, $z = -11.38$, $p < .001$, $r = -.35$. On the other hand, the amount of virtual contact was significantly lower at follow-up than at baseline, $z = -9.15$, $p < .001$, $r = -.28$. S5 Table in S1 Appendix presents cross-country differences in contact with loved ones and time spent at home at T1 and T2.

**Financial situation.** At T1, 34.6% of participants ($n = 370$) reported a pandemic-related income loss, whereas this was the case for only 22.8% ($n = 244$) at T2, a statistically significant difference $\chi^2(1) = 65.65$, $p < .001$, OR = 0.31. This reduction resulted from 182 people (17%) reporting an income loss only at T1 and 56 people (5.2%) reporting an income loss only at T2.

Of $n = 370$ participants who reported a pandemic-related income loss at baseline, only $n = 45$ (12.2%) received financial support from the government. At follow-up, the percentage of affected people receiving financial support decreased to 9% ($n = 22$); this difference was not statistically significant: $\chi^2(1) = 3.45$, $p = .06$, OR = 0.45. Details on pandemic-related income reduction and financial support can be found in S6 Table in S1 Appendix.

## Quantitative results

**Mental health changes over time.** Paired-samples $t$-tests on the total sample showed a significant decrease in symptoms of depression over time but no significant difference in symptoms of AD, PTSD, and well-being (for details, see Table 2). At the country level, symptoms of depression decreased in Croatia and Greece while symptoms of AD increased in Georgia and decreased in Greece. Due to a very low number of participants with trauma exposure per country, $t$-tests could not be conducted for the PC-PTSD-5. Therefore, only mean values on this scale for each country were calculated.

**Mental health changes across countries.** At baseline, symptoms of AD differed significantly between countries, Welch's $F(4, 296.499) = 18.712$, $p < .001$, est. $\omega^2 = .53$, with Georgia,

**Table 2. Means, standard deviations, and *t*-test statistics for mental health outcomes.**

| | T1 | T2 | | | |
|---|---|---|---|---|---|
| | *M (SD)* | *M (SD)* | *t(df)* | *p* | Cohen's *d* |
| | | | ADNM-8 | | |
| Austria | 14.88 (5.78) | 14.83 (6.14) | 0.16 (332) | .873 | 0.009 |
| Croatia | 13.80 (4.82) | 14.02 (4.91) | -1.06 (413) | .292 | 0.052 |
| Georgia | 18.53 (6.84) | 19.69 (7.21) | -2.13 (112) | **.035** | 0.200 |
| Greece | 16.85 (6.20) | 15.47 (5.96) | 2.75 (121) | **.007** | 0.249 |
| Portugal | 17.15 (6.64) | 16.53 (6.24) | 1.10 (87) | .274 | 0.117 |
| Total sample | 15.26 (5.90) | 15.24 (6.04) | 0.11 (1069) | .911 | 0.004 |
| | | | PHQ-2 | | |
| Austria | 1.29 (1.41) | 1.20 (1.42) | 1.11 (332) | .267 | 0.061 |
| Croatia | 1.27 (1.43) | 1.12 (1.50) | 2.16 (413) | **.032** | 0.106 |
| Georgia | 1.72 (1.78) | 1.62 (1.93) | 0.52 (112) | .604 | 0.049 |
| Greece | 1.61 (1.47) | 1.11 (1.34) | 3.49 (121) | **.001** | 0.316 |
| Portugal | 1.80 (1.90) | 1.70 (1.68) | 0.51 (87) | .614 | 0.054 |
| Total sample | 1.40 (1.52) | 1.24 (1.54) | 3.41 (1069) | **.001** | 0.104 |
| | | | WHO-5 | | |
| Austria | 58.21 (22.23) | 56.60 (23.35) | 1.59 (332) | .113 | 0.087 |
| Croatia | 58.63 (20.73) | 57.64 (20.44) | 1.19 (413) | .235 | 0.058 |
| Georgia | 49.95 (22.22) | 46.65 (22.00) | 1.44 (112) | .144 | 0.135 |
| Greece | 52.49 (20.11) | 54.43 (19.35) | -1.13 (121) | .262 | 0.102 |
| Portugal | 53.82 (20.46) | 53.82 (21.88) | 0.00 (87) | .999 | 0.000 |
| Total sample | 56.49 (21.47) | 55.48 (21.76) | 1.77 (1069) | .077 | 0.054 |
| | | | PC-PTSD-5 [a] | | |
| Austria [b] | 1.38 (1.26) | 1.38 (1.49) | | | |
| Croatia [c] | 1.74 (1.51) | 1.37 (1.54) | | | |
| Georgia [d] | 2.83 (2.04) | 3.00 (1.86) | | | |
| Greece [e] | 1.50 (1.08) | 1.00 (1.29) | | | |
| Portugal [f] | 2.50 (0.70) | 2.67 (1.94) | | | |
| Total sample [g] | 1.85 (1.54) | 1.59 (1.52) | 1.93 (94) | .057 | 0.198 |

ADNM-8 = Adjustment Disorder New Module-8; PC-PTSD = Primary Care PTSD Screen for DSM-5; PHQ-2 = Patient Health Questionnaire-2; WHO-5 = 5-item World Health Organization Well-Being Index. Significant differences are written in bold. Austria: *n* = 333. Croatia: *n* = 414. Georgia: *n* = 113. Greece: *n* = 122. Portugal: *n* = 88.

[a] Only participants who reported having experienced a traumatic event during the pandemic were included in the analysis.

[b] T1: *n* = 13; T2: *n* = 64.

[c] T1: *n* = 143; T2: *n* = 130.

[d] T1: *n* = 6; T2: *n* = 16.

[e] T1: *n* = 10; T2: *n* = 19.

[f] T1: *n* = 2; T2: *n* = 9.

[g] *n* = 95.

Greece, and Portugal having higher ADNM-8 scores than Austria ($p \leq .033$) and Croatia ($p < .001$). Depressive symptoms also differed between the countries, Welch's $F(4, 300.714) = 3.760$, $p = .005$, est. $\omega^2 = .53$, though no significant pairwise comparisons were found. Differences between the five countries were also found on the WHO-5, $F(4, 1065) = 5.683$, $p < .001$, $\omega^2 = .02$. Tukey post hoc analysis revealed significantly lower well-being in Georgia than in Austria ($p = .003$) and Croatia ($p = .001$), and lower well-being in Greece than in Croatia ($p = .042$).

At follow-up, rates of AD symptoms remained significantly different between the countries, Welch's $F(4, 299.216) = 17.469$, $p < .001$, est. $\omega^2 = .53$. The post hoc analysis revealed higher ADNM-8 scores in Georgia than in all other countries ($p \leq .001$) and significantly higher scores in Portugal than in Croatia ($p = .005$). Scores for depression also differed significantly at follow-up, Welch's $F(4, 305.864) = 3.751$, $p = .005$, est. $\omega^2 = .53$, with higher PHQ-2 scores in Portugal than in Croatia ($p = .024$) and Greece ($p = .048$). Moreover, the WHO-5 scores differed significantly between the countries at T2, Welch's $F(4, 312.703) = 6.075$, $p < .001$, est. $\omega^2 = .02$. Again, lower well-being was found in Georgia compared to Austria ($p = .007$) and Croatia ($p = .002$), and in Greece compared to Croatia ($p = .003$). No ANOVA was conducted for the PC-PTSD-5 due to small and unequal subsample sizes.

**Prevalence rates of probable mental health disorders.** In the total sample, the percentage of people at risk for developing AD or PTSD did not change over time. However, the prevalence rates of self-reported probable depression decreased significantly from T1 to T2 ($p = .027$). All results, including within country comparisons, are depicted in Table 3. For PTSD, only descriptive statistics are shown due to low number of participants reporting a traumatic event during the pandemic (Austria: $n_{t1} = 13$, $n_{t2} = 64$; Croatia: $n_{t1} = 143$, $n_{t2} = 130$; Georgia: $n_{t1} = 6$, $n_{t2} = 16$; Greece: $n_{t1} = 10$, $n_{t2} = 19$; Portugal: $n_{t1} = 2$, $n_{t2} = 9$).

## Qualitative results

The themes identified at T1 proved to be well-suited for coding the participants' answers at T2. However, for each of the four questions, one additional theme regarding vaccination emerged. The final list of themes and categories, including definitions and anchor examples, is presented in S2 Appendix. Figs 1–4 illustrate the differences in the frequency of themes over time and across countries (for exact values, see S3 Appendix).

In the following, we touch upon the most prominent differences, thereby using semi-quantification (i.e. simple counts; see Table 4). We also present selected quotes from T2 (with reference to gender, age, and country of residence), which reflect the diversity of the sample, strong patterns found in the data, and country-specific characteristics. An extensive overview of themes and quotes identified at T1 is provided elsewhere [62].

**The most stressful event.** *Restrictions and changes in daily life* was one of three most prominent themes in all countries at both time points, and comprised different challenges experienced by the participants:

*"My grandson was born in Germany. Now he's six months old and we still haven't touched each other." (CRO, male, 55)*

*"The population does not have the basic opportunity to use public transport. All public institutions are working except transport and I think it's dumb to restrict transport at this time." (GEO, female, 19)*

*"Not being able to have a normal life." (PT, male, 67)*

Over time, the frequency of answers related to restrictions and changes decreased in Austria and Georgia. Meanwhile, answers pertaining to the theme *COVID-19 and other health issues* increased in all countries, with COVID-19-related deaths and (long-term) symptoms being commonly reported:

*"Disease, I mean having COVID-19, or better said, the symptoms of the disease, especially the high temperature for 12 days!" (CRO, male, 45)*

**Table 3. Prevalence rates of probable mental disorders at T1 and T2.**

| | T1 | T2 | | | |
|---|---|---|---|---|---|
| | *n* (%) | *n* (%) | $\chi^2(1)$ | *p* | OR |
| **ADNM-8** | | | | | |
| Austria | 46 (13.8) | 45 (13.5) | 0.00 | .999 | 0.96 |
| Croatia | 24 (5.8) | 33 (8.0) | 1.94 | .164 | 1.75 |
| Georgia | 30 (26.5) | 40 (35.4) | - [a] | .052 | 2.66 |
| Greece | 24 (19.7) | 15 (12.3) | 2.37 | .124 | 0.50 |
| Portugal | 24 (27.3) | 18 (20.5) | - [b] | .238 | 0.50 |
| Total sample | 148 (13.8) | 151 (14.1) | 0.03 | .871 | 1.04 |
| **PC-PTSD-5** [e] | | | | | |
| Austria | 0 (0.0) | 7 (10.9) | | | |
| Croatia | 23 (16.1) | 15 (11.5) | | | |
| Georgia | 2 (33.3) | 9 (56.3) | | | |
| Greece | 0 (0.0) | 1 (5.3) | | | |
| Portugal | 0 (0.0) | 5 (55.6) | | | |
| Total sample | 25 (14.4) | 37 (15.5) | | | |
| **PHQ-2** | | | | | |
| Austria | 48 (14.4) | 41 (12.3) | 0.66 | .418 | 0.77 |
| Croatia | 63 (15.2) | 56 (13.5) | 0.52 | .470 | 0.82 |
| Georgia | 26 (23.0) | 25 (22.1) | - [c] | .999 | 0.92 |
| Greece | 28 (23.0) | 15 (12.3) | 5.33 | **.021** | 0.35 |
| Portugal | 23 (26.1) | 19 (21.6) | - [d] | .503 | 0.66 |
| Total sample | 188 (17.6) | 156 (14.6) | 4.90 | **.027** | 0.72 |

OR = Odds ratio; ADNM-8 = Adjustment Disorder New Module-8; PC-PTSD = Primary Care PTSD Screen for DSM-5; PHQ-2 = Patient Health Questionnaire-2. The calculation of prevalence rates was based on the established cut-off scores: ADNM-8 > 22, PC-PTSD-5 > 3, and PHQ-2 > 2. Significant differences are written in bold. Binomial distribution was used if the sum of the observed counts (i.e., discordant pairs) was *n* < 26:

[a] *n* = 22.

[b] *n* = 18.

[c] *n* = 25.

[d] *n* = 20.

[e] Only persons who reported a traumatic event during the pandemic were included in the analysis (T1: *n* = 174, T2: *n* = 238).

> *"Death of my brother and several good friends due to corona"* (AT, female, 54)

> *"Being on sick leave frequently because of self-isolation and post-covid symptoms, and the recovery from COVID (long-term emotional sensitivity and migraines)"* (CRO, female, 42)

*Emotional distress* was frequently mentioned at both time points. However, a marginal decrease in the frequency of this theme was observed at T2 in all countries. As illustrated by the quotes below, fear and uncertainty were common stressors:

> *"Fear of losing my job. Fear that I will lose all the people I love. Fear that my life will never again be good and meaningful. Fear that I will not know how to run a private company and that I will become poor. I am afraid of poverty, although I have no objective reasons for that–at least not for now."* (CRO, female, 52)

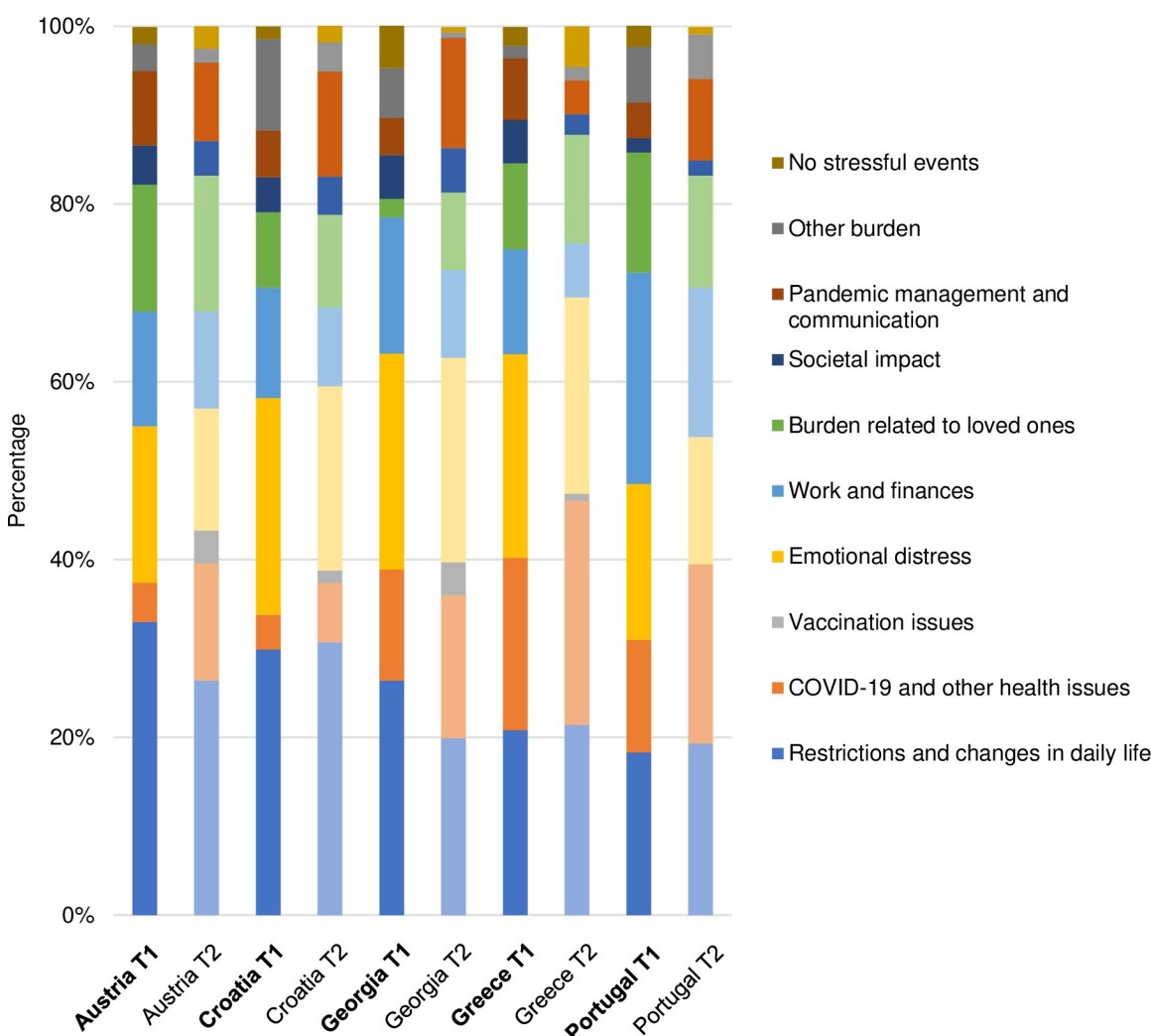

**Fig 1. Themes related to stressful events during the pandemic.**

*"The unpredictability of the future was very worrying. Dealing with the fear of other people (family members and co-workers) was also stressful"* (PT, female, 32)

*"The openings in June 2020. Neither vaccinated nor recovered, I felt very uncomfortable participating in social activities again and I largely withdrew. Not necessarily out of fear of getting infected myself but of infecting someone else."* (AT, male, 29)

The theme *Work and finances*, which showed a downward trend over time in all countries, included answers about high workload, job losses, and unemployment. Financial burden was particularly pronounced in Portugal, for instance:

*"Being forced to work overtime hours and it takes months to pay them, and I still don't know if I'm going to get paid in full; being located far from my place of work and residence and not being paid for travel. I am still at a loss and the situation is still ongoing and will continue."* (PT, female, 52)

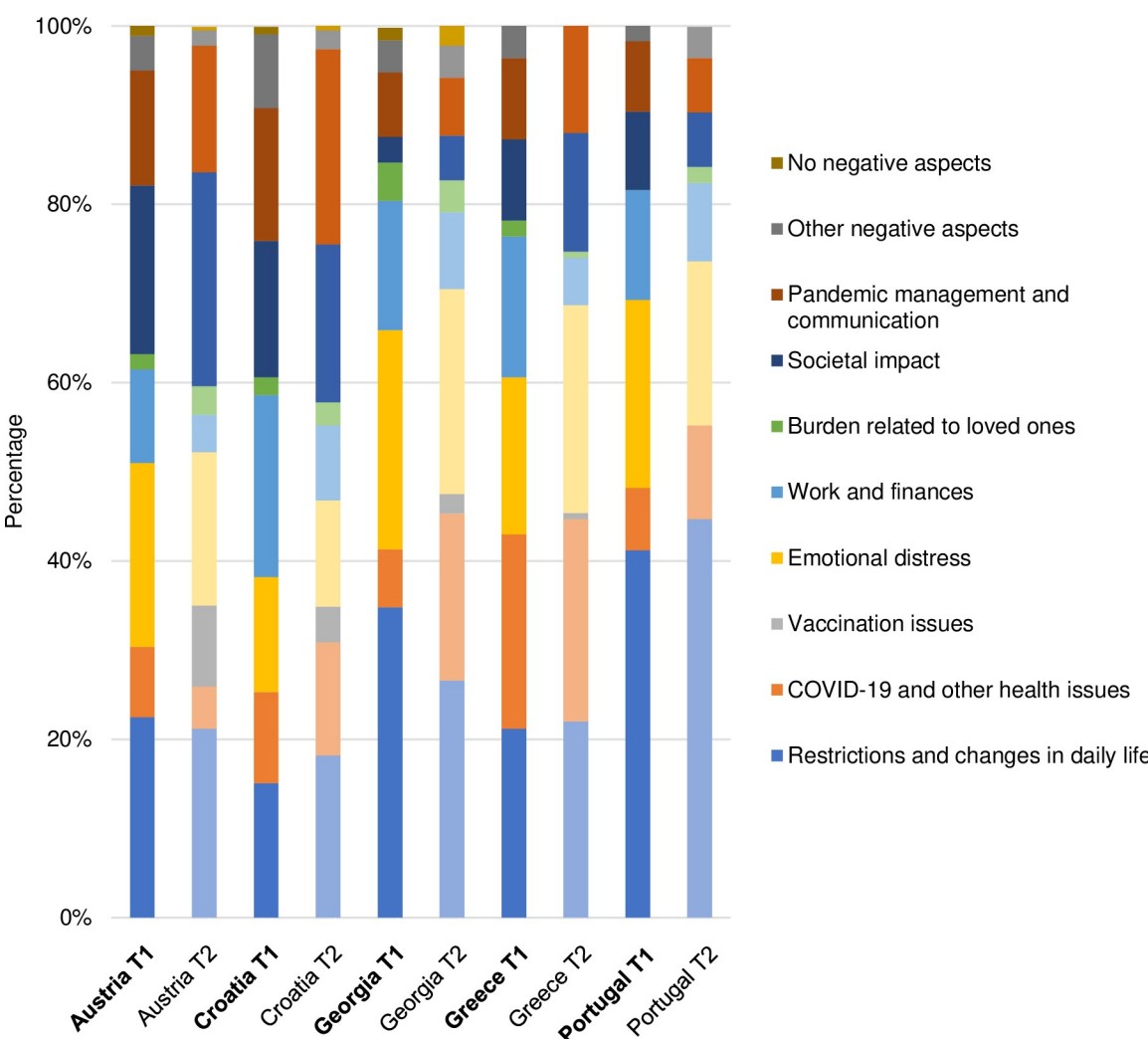

**Fig 2. Themes related to negative aspects of the pandemic.**

For the theme *Burden related to loved ones*, a small increase between T1 and T2 was observed in all countries except for Portugal. Common answers included the death of loved ones and problems with childcare, e.g.:

*"My husband's grandfather died and we had not seen him for a year because of the pandemic." (CRO, female, 34)*

*"Childcare, at the same time learning [home-schooling] with two children (aged 11 and 14) on days off and then going back to work in health care (shift work)." (AT, male, 44)*

For the theme *Pandemic management and communication*, different patterns of change were observed between the countries. Whereas the frequency of this theme increased in Croatia, Georgia, and Portugal, it decreased in Greece. For Austria, no notable differences were found.

*"Lack of reassuring information from the sovereign bodies to the public." (PT, female, 55)*

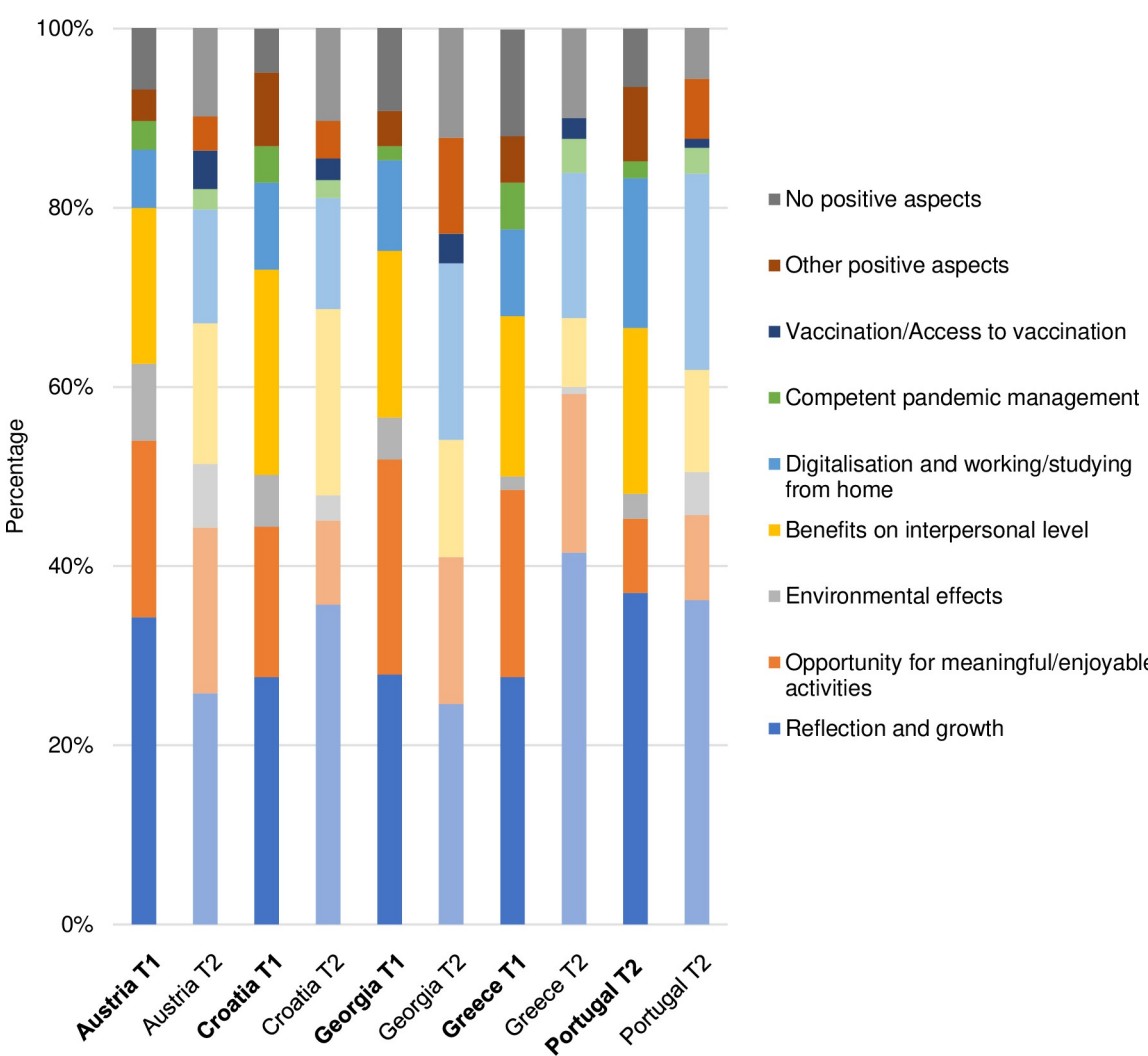

**Fig 3. Themes related to positive aspects of the pandemic.**

*"Contradictory information from the authorities, idiocy of local governments and particularly contradictory decisions of the authorities." (CRO, male, 48)*

*Vaccination issues* were identified as a theme only at T2. This theme was most often reported in Austria and Georgia (3.7%) and was not mentioned in Portugal.

*"There is vaccination, and many don't go [to get vaccinated]." (AT, male, 36)*

*"Myths about vaccination. My relatives whined that I won't be able to have kids [after being vaccinated]." (GEO, female, 24)*

At times, multiple stressful events co-occurred, resulting in a significant burden for the participants, as illustrated below:

*"My dad could not have an urgent surgery because he got infected during a medical exam in hospital. Due to postponing the surgery, his symptoms severely deteriorated. He has been in*

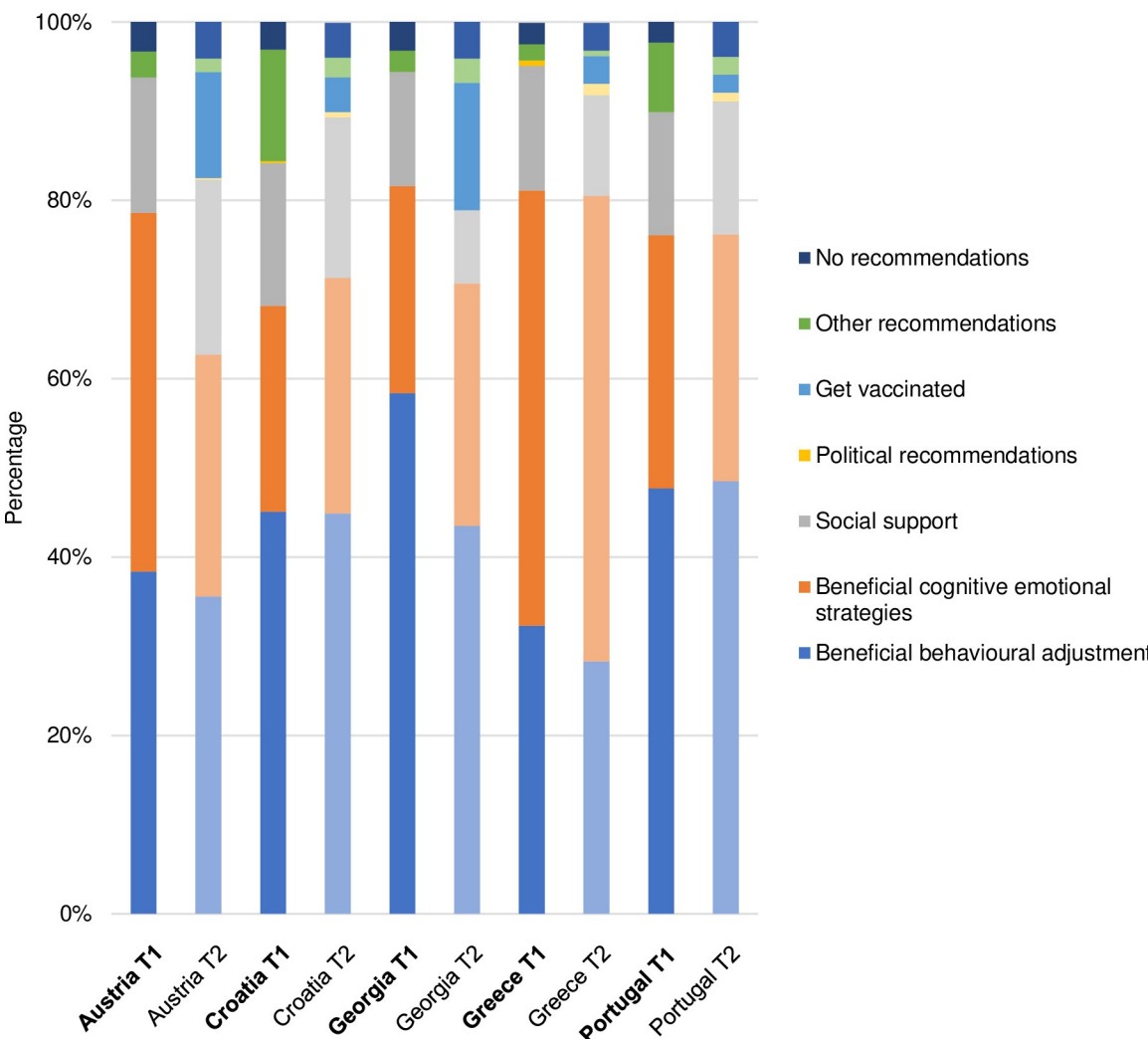

**Fig 4. Themes related to recommendations how to cope with the pandemic.**

*hospital for six months, he cannot move anymore, because of the tracheostomy we could not even communicate with him over the phone. And because of the measures we cannot visit him. Medical doctors are overworked, it is hard to get the information you need. Being on the phone with the hospital and uncertainty are a huge mental health burden for all of us. Mom is in a different city, alone, after living with dad for 50 years. And her life has changed. And I am really worried every day because of their future, which will be completely different than before." (CRO, female, 47)*

**Negative aspects of the pandemic.**   Similar to the first question, *Restrictions and changes in daily life* also proved to be a common theme in response to the second question about negative aspects of the pandemic, both at T1 and T2. Between the two time points, a clear downward trend was observed in Georgia. In other countries, the frequencies barely changed over time, with Portuguese participants most often describing restrictions and pandemic-related changes as negative:

**Table 4. Identified themes and their frequencies at baseline and follow-up.**

| Theme | T1 | T2 |
|---|---|---|
| **The most stressful event** | | |
| Restrictions and changes in daily life | 51.6% | 48.4% |
| COVID-19 and other health issues | 36.5% | 63.5% |
| Vaccination issues | 0 | 100.0% |
| Emotional distress | 53.5% | 46.5% |
| Work and finances | 57.6% | 42.4% |
| Burden related to loved ones | 45.4% | 54.6% |
| Societal impact | 50.9% | 49.1% |
| Pandemic management and communication | 37.8% | 62.2% |
| Other burden | 71.8% | 28.2% |
| No stressful events | 49.2% | 50.8% |
| **Negative aspects of the pandemic** | | |
| Restrictions and changes in daily life | 49.5% | 50.5% |
| COVID-19 and other health issues | 46.7% | 53.3% |
| Vaccination issues | 0 | 100.0% |
| Emotional distress | 51.8% | 48.2% |
| Work and finances | 69.4% | 30.6% |
| Burden related to loved ones | 42.6% | 57.4% |
| Societal impact | 44.9% | 55.1% |
| Pandemic management and communication | 43.8% | 56.3% |
| Other negative aspects | 72.7% | 27.3% |
| No negative aspects | 60.0% | 40.0% |
| **Positive aspects of the pandemic** | | |
| Reflection and growth | 50.4% | 49.6% |
| Opportunity for meaningful/enjoyable activities | 58.4% | 41.6% |
| Environmental effects | 62.2% | 37.8% |
| Benefits on interpersonal level | 56.6% | 43.4% |
| Digitalisation and working/studying from home | 40.8% | 59.2% |
| Competent pandemic management | 63.5% | 36.5% |
| Vaccination/Access to vaccination | 0 | 100.0% |
| Other positive aspects | 58.8% | 41.2% |
| No positive aspects | 42.1% | 57.9% |
| **Recommendations for dealing with the pandemic** | | |
| Beneficial behavioural adjustment | 51.5% | 48.5% |
| Beneficial cognitive-emotional strategies | 51.7% | 48.3% |
| Social support | 47.4% | 52.6% |
| Political recommendations | 22.2% | 77.8% |
| Get vaccinated | 0 | 100.0% |
| Other recommendations | 78.7% | 21.3% |
| No recommendations | 43.9% | 56.1% |

*"The loss of routine and opportunities derived from regular contact with colleagues and superiors at work."* (PT, male, 35)

*"Loss of social skills due to mandatory isolation; loss of physical contact with people beyond the immediate household—spontaneous hugs, for example."* (PT, female, 59)

For the theme *COVID-19 and other health issues*, several differences were observed across countries and between time points. The frequency of this theme was rather low in Austria, Croatia, and Portugal, with only marginal changes between T1 and T2. In Georgia and Greece, approximately one in five participants commented on deaths and health issues related to COVID-19, for instance:

> *"People are dying and in my country their lives are not important." (GEO, female, 34)*

> *"Loss of sense of taste and smell for a long time (about 1 month) during infection." (GEO, male, 22)*

> *"Too many people perished too quickly and painfully" (GR, female, 40)*

The theme *Emotional distress* was also represented differently depending on the country. Over time, the number of answers pertaining to this theme decreased in Austria and Portugal. At follow-up, a high number of emotional difficulties was still evident in Greece and Georgia, mostly involving the widespread uncertainty and anxiety:

> *"Constant uncertainty about the future." (GEO, female, 19)*

> *"Anxiety about the next minute." (GR, female, 40)*

In all countries, the theme *Work and finances* showed a downward trend between T1 and T2, with the sharpest decrease in Croatia. At follow-up, this theme was most prominent in Portugal and Georgia.

> "*The destruction of so many jobs and the economy in general" (PT, female, 52)*

> *"In the long run, I am concerned about the economic costs of the pandemic and its reflection on our country's situation" (PT, female, 54)*

> *"Losing my job. I . . . worked [in a school] for only 2 months, then Corona began and I could not do anything, so I lost my job. I have a grandmother with breast cancer operation, I have 5000 loan in bank and many other loans. . . . I'm paying these loans every day and can't find a job. If the lockdown begins again, we'll stay hungry, we don't even have any gold to take it to pawnshop." (GEO, female, 29)*

The theme *Societal impact* was more prominent at T2 than T1 in all countries except for Portugal. Particularly high number of participants commenting on the reactions of the population and societal changes was observed in Austria and Croatia, e.g.:

> *"The impossibility for children and adolescents to be allowed to live their lives in a manner appropriate for their age, to know they are on safe paths, and to be able to make joyful plans for the future." (AT, female, 73)*

> *"The fact that once again the part of society that is marginalised, poor and in the lowest social classes suffered the most damage, while the rich became even richer." (CRO, female, 25)*

> *"Increased segregation in coping with the pandemic between rich countries and those at a lower level of development." (CRO, male, 25)*

For the theme *Pandemic management and communication*, rather small changes were observed over time. This was true for all countries except Croatia, where a notable increase was found, as illustrated below:

*"In the media, the spread of lies, hatred, stupidity, corruption, malicious reinterpretation, shortening and changing of the statements of doctors and other experts and politicians."* (CRO, male, 45)

*"The role of the media. The way the leading structures talk about the pandemic in the media, spreading fear, fuelling anxiety. Inconsistent ways of controlling the pandemic (unequal measures and conditions for everyone, illogical nature of measures). Not focusing on mental health (until it's (too) late)."* (CRO, female, 32)

*Burden related to loved ones* was rarely reported, with only negligible differences between baseline and follow-up in all countries.

*Vaccination issues* emerged as a theme at follow-up in all countries except for Portugal. Statements regarding (mandatory) vaccination, including population's reactions towards it, were most common in Austria, followed by Croatia.

*"Mandatory vaccination through the so-called back door with the regulations. I strongly refuse compulsory vaccination, especially for children and young people."* (AT, male, 71)

*"The fact that despite the population's great willingness to be vaccinated, there are still very many anti-vaxxers."* (AT, female, 72)

*"Even those who do not belong to any vulnerable group (or have even recovered from Covid) get vaccinated and are ready to vaccinate their own children with an experimental genetic preparation . . . Apart from the possible negative health consequences, all this can easily turn into a path towards a totalitarian society."* (CRO, male, 39)

*"Pressure on people to get vaccinated, threats from the government about what will happen if we don't get vaccinated. . ."* (CRO, female, 55)

**Positive aspects of the pandemic.** *Reflection and growth* constituted the most prominent positive theme in all countries at both time points. Over time, the number of people reporting individual and community growth (e.g. better awareness of certain topics, reprioritisation and lessons from COVID-19) decreased substantially in Austria, remained largely unchanged in Portugal and Georgia, and increased in Croatia and Greece. Responses regarding reflection and growth included:

*"Learning about the fragility of human existence, facing the fact that despite scientific advances we are not omnipotent, facing limitations that we must learn to deal with and be aware of."* (CRO, male, 79)

*"Changing the perspective on life. A situation that encouraged many to think about priorities and wishes."* (CRO, female, 32)

*"The rethinking of the values in our lives and that we were shaken up about the things we took for granted, like being alive and having a job."* (GR, male, 41)

*"The appreciation of health and how important the people we love are, and we might lose them overnight."* (GR, female, 31)

*Opportunity for meaningful/enjoyable activities* was another common theme. However, over time, the frequency of this theme tended to decrease in all countries except for Portugal, where a negligible increase was observed. Answers relating to having more time and doing

pleasurable activities were particularly widespread among Austrian and Greek participants, for instance:

> *"That it became quieter at times on the streets, but also in everyday life (fewer appointments)." (AT, female, 40)*

> *"More time for yourself. Deceleration: -)" (AT, male, 55)*

> *"Time with myself for myself" (GR, female, 53)*

> *"More time with my family" (GR, female, 29)*

At baseline, *Benefits on the interpersonal level* represented a recurrent theme in all countries, with a large number of people positively evaluating family bonding and social cohesion during the pandemic. Notably, in all countries, the frequency of this theme tended to decline over time. At follow-up, benefits on the interpersonal level were mainly reported by participants from Austria and Croatia, e.g.:

> *"That many people found each other again, and that family bonding was strengthened." (AT, male, 81)*

> *"People have started to socialise at home again, instead of in cafes. The pressure to participate in social activities and to be constantly active and visible has decreased." (CRO, female, 40)*

> *"I also find the reduced amount of contact to be a positive thing because it seems to me that the relationships with irrelevant people partly or completely disappeared, whereas the relationships with loved ones/close friends have deepened because we spent more time together." (CRO, female, 23)*

In contrast to the themes described above, the theme *Digitalisation and working/studying from home* was more prominent at T2 than at T1. The frequencies were constantly highest in Portugal, whereas the sharpest increase between T1 and T2 was noted in Georgia.

> *"Opportunity to study online, I'm glad that I don't waste time on transport anymore, I have a lecture when I want, [. . .] I can attend from where I want to." (GEO, female, 24)*

> *"Working from home." (GEO, male, 22)*

> *"Better organisation of almost all services, public or private (e.g. restaurants), including greater provision of delivery services." (PT, male, 35)*

> *"The rise of telework as a topic on the agenda in social issues, and how important it can be for the positive performance of the worker, if correctly applied." (PT, male, 38)*

Positive aspects pertaining to *Competent pandemic management* (e.g. competence of health workers) and *Environmental effects* (e.g. less pollution) were less prominent, with a downward trend over time in all countries except for Portugal. Remarkably, the number of people who did not identify anything positive about the pandemic increased over time in Austria, Croatia, and Georgia.

**Recommendations for dealing with the pandemic.** *Beneficial behavioural adjustment* was the most prominent theme across countries and at both time points. Recommendations to adhere to preventive measures, engage in pleasant activities, and inform oneself adequately were almost equally represented at T1 and T2. Over time, a notable decrease in frequency of

this theme was observed only in Georgia. Some of the participants' recommendations were as follows:

> *"Stay informed via official channels, follow the recommendations of professional services."* (CRO, male, 45)

> *"Be careful, follow the measures, but do not adhere to them blindly. Find some time and space for what makes you happy, even if it is a short bike ride or a documentary film about a foreign country or animals."* (CRO, female, 47)

> *"Stay active, interested in living everyday life as normally as possible."* (PT, female, 55)

> *"Physical activity, activities that promote well-being and relaxation, adequate rest time, etc."* (PT, male, 39)

*Beneficial cognitive-emotional strategies* represented the second most prominent theme and showed different patterns of change across the countries. The frequency of this theme substantially decreased in Austria and slightly increased in Croatia, Georgia, and Greece, while being constantly high in Portugal. Recommendations pertaining to this theme included:

> *"Stay calm and be patient and everything will get better."* (GR, female, 22)

> *"Know that no matter how difficult this whole situation has been or still is for them, it will end and normality will return."* (GR, male, 41)

> *"Calm down, take a deep breath, accept the reality."* (PT, female, 54)

> *"Focus on things that make you happy, think positively, set goals and never give up."* (GEO, female, 20)

Recommendations referring to the theme of *Social support* included keeping in contact with other people and seeking support when needed. Whereas such recommendations were more common at T2 than at T1 in Austria, Croatia, and Portugal, the opposite was true in Greece and Georgia.

> *"Pay attention to fellow human beings and refrain from selfishness."* (AT, male, 44)

> *"Seek any form of help (physician, psychologist, psychiatrist. . .) in good time."* (CRO, male, 52)

> *"Surround yourselves with people you can talk to, even if only online."* (PT, female, 31)

A newly emerging recommendation theme at follow-up was *Get vaccinated*. This theme came up in all countries, but was particularly prevalent in Austria and Georgia, e.g.:

> *"Get vaccinated on time."* (GEO, female, 55)

> *"Get vaccinated so that we can get the pandemic under control."* (AT, female, 68)

The themes *Political recommendations* and *No recommendations* were rarely reported at either time point but showed an upward trend over time. Specifically, there was an increase in the number of participants providing recommendations of a political nature (e.g. "Go vote and oust this government") and participants not providing any recommendations (e.g. "I don't have any recommendations because I don't even know how to help myself").

## Pandemic-related experiences by group

In the following, the most important results of the mixed-methods analysis are described. (S5-S12 Tables in S3 Appendix) contains comprehensive comparisons of themes based on sociodemographic and health-related characteristics, social factors, and financial situation of the participants.

The integration of qualitative and quantitative data using revealed that *Restrictions and changes in daily life* were more often reported as stressful and negative by people who were single and people living alone. Women and people with children more frequently expressed *Burden related to loved ones*, whereas men were more likely to report *Pandemic management and communication* as stressful and were less likely to report any negative aspect of the pandemic. Notably, participants with better self-reported health tended to perceive *Vaccination issues* as stressful and negative, whereas those with less face-to-face contact with loved ones at baseline tended to indicate *Emotional distress* and *Work and finances* as stressful.

With respect to positive aspects of the pandemic, *Benefits on the interpersonal level* were more often reported by female participants, by participants in a relationship, and by those with children. On the contrary, participants without children predominantly reported *Opportunity for meaningful/enjoyable activities*. Furthermore, participants with less face-to-face contact were more likely to perceive *Digitalisation and working/studying from home* as positive, while those with more face-to-face contact were more likely to report *Environmental effects* as a positive aspect of the pandemic.

Notably, women and participants without children mainly provided recommendations pertaining to the theme of *Social support*. Participants who did not provide any recommendations had, on average, slightly worse self-reported health, less face-to-face contact with loved ones, and often experienced a pandemic-related income loss.

In terms of age differences, we found a higher mean age for participants whose responses to the question about negative aspects included the themes *COVID-19 and other health issues* ($M_{age}$ = 49.0) and for participants whose answers to the question about stressful events included the themes *Pandemic management and communication* ($M_{age}$ = 46.6) and *No stressful events* ($M_{age}$ = 46.3). *Opportunity for meaningful/enjoyable activities* was a common positive aspect of the pandemic for somewhat younger participants ($M_{age}$ = 39.6) while *Competent pandemic management* ($M_{age}$ = 50.1) and *Vaccination/Access to vaccination* ($M_{age}$ = 55.5) were commonly reported by older participants. Concerning recommendations, *Get vaccinated* was mostly recommended by older participants ($M_{age}$ = 49.8) and *Political recommendations* were articulated mainly by younger participants ($M_{age}$ = 39.4).

## Discussion

The present study investigated subjective pandemic-related experiences over time and across countries. Using a mixed-methods approach, we aimed to gain a better understanding of the factors underlying changes in mental health between the first and second year of the COVID-19 pandemic. Our analyses revealed significantly different mental health outcomes in Austria, Croatia, Georgia, Greece, and Portugal at both time points. Furthermore, we identified changes in subjective experiences which mirrored the shifting context of the pandemic. By and large, our findings indicate that people's reactions to the pandemic vary considerably, highlighting the role of individual and country-specific factors in mental health research and practice peri- and post-COVID-19.

### Mental health differences over time and across countries

In line with our hypothesis and the current literature, different mental health outcomes showed different trajectories over time. At the cross-country level (i.e. overall sample),

depressive symptoms decreased between T1 (i.e. summer and autumn 2020) and T2 (i.e. summer and autumn 2021), thus corroborating the results found in the USA in a similar period [13]. We did not find any significant differences regarding AD, PTSD, and well-being scores in the overall sample.

At the country level, significant decreases in depressive symptoms were observed in Croatia and Greece. These two countries had the highest percentages of people without a current or previous diagnosis of a mental disorder, potentially indicating a generally better mental health status than in the other three countries. This may have fostered a better adjustment to the pandemic and a faster recovery. The observed reduction of depressive symptoms in Croatia might also be attributed to the very high percentage of parents in this subsample, as a strengthening of family relationships and more time spent with family members are commonly reported positive consequences of the pandemic for parents [67], which may in turn protect their mental health.

In Greece, we observed a decrease in AD symptoms. Even though Greece has been hit hard by the pandemic and government measures were stricter than in other countries (as illustrated in S1 Appendix), people seem to have found a way to accept the situation and adjust to it over time. This difference between Greece and the other countries might be related to mental toughness, which was shown to be higher in Greece than in some other European countries [68]. Mentally tough individuals tend to perceive challenge as an opportunity for personal growth and respond adaptively to stressors [69]. As can be seen in the qualitative data, a large proportion of Greek participants described gratitude and appreciation, as well as a restructuring of values and priorities, as positive aspects of the pandemic. They also repeatedly emphasised the importance of calmness, patience, and optimism for coping with the pandemic. Many of these aspects (i.e. values clarification, gratitude, positive thinking) have been identified as protective against mental health problems [36, 70]. They have also shown promising effects in interventions addressing COVID-19-related distress by targeting psychological flexibility [71]. Furthermore, psychological flexibility might be essential for explaining psychological adjustment to the pandemic [39].

Notably, symptoms of AD increased significantly in Georgia. In this country, we also found a higher prevalence of self-reported AD at T2 (35.4%) than at T1 (26.5%). Though being only nearly significant, this finding, along with the observed increase in symptoms, suggests difficulties in adjusting to the pandemic among the Georgian population. This might be explained by the fact that the Georgian subsample was the youngest and had the highest proportion of female participants. Pandemic-related income loss was also highest in Georgia, with 40.7% and 50.4% of participants being affected at T2 and T1, respectively. It is known that young people, women, and people with pre-existing financial difficulties have been particularly burdened by the pandemic [8, 12], and all three of these groups were overrepresented in the Georgian subsample. Moreover, Georgia had the lowest COVID-19 vaccination rate and the highest incidence and death rate of the five countries at follow-up, which may also have impeded their adjustment to the pandemic. The increase in the frequency of themes related to COVID-19 and pandemic management further supports this assumption. Many Georgian participants indicated being burdened by COVID-related deaths, vaccination issues, and inadequate healthcare in their country.

In Austria and Portugal, none of the assessed mental health outcomes changed significantly over time. These two countries showed similarities regarding living situation and relationship status and also had a higher mean age than the other countries. The latter might have helped them to maintain stable mental health over time, as older people have been shown to adapt better to containment measures [72]. Another protective factor might be the comparably low percentage of participants reporting a pandemic-related income loss in these two countries.

The relatively stable financial situation might have protected Austrian and Portuguese participants from further loss spirals and associated negative effects on mental health, as proposed by the COR theory [35]. It is also noteworthy that Austria and Portugal have the highest GDP per capita among the participating countries [73] and that more favourable outcomes in these two countries were already demonstrated in an earlier international study [31]. Thus, it seems plausible that both pre-existing advantages and national responses to the pandemic contribute to cross-country differences in mental health.

With regard to the symptoms of PTSD, only descriptive analysis could be conducted due to low and unequal sample sizes. Nevertheless, in all countries except from Croatia, we observed an increase in the number of people who experienced a traumatic event during the pandemic. In Croatia, two major earthquakes happened in 2020, leading to a double adversity and an increase in demand for mental health services [74]. This might explain a higher number of Croatian participants reporting trauma exposure at T1 than at T2. The increase of traumatic events in the remaining countries underlines the need for future research on posttraumatic symptoms in individuals exposed to trauma during COVID-19.

When interpreting the changes over time, it is important to consider that all mental health outcomes differed significantly between the five countries at both time points. In fact, over 50% of the total variance in adjustment and depression scores was attributable to country, highlighting the role of country-specific factors in understanding the mental health impact of COVID-19. Multiple studies have already reported cross-country differences in mental health amidst the pandemic [22]. These differences seem to be associated with the strictness of governmental restrictions [75] and variations in vaccine acceptance [76], but also with standard of living [77] and quality of healthcare [78]. All of these aspects are likely to have impacted our results, considering the variety of socioeconomic and cultural factors across the countries examined.

## Pandemic-related experiences

In accordance with our assumptions, pandemic-related experiences changed over time. In the following, we touch on the main differences in participants' experiences over time and across countries.

**Stressful events and negative aspects of the pandemic.** In the questions about stressful events and negative aspects of the pandemic, one new theme relating to vaccination emerged: Participants expressed concerns about side effects of the vaccination and dissatisfaction regarding the pressure to get vaccinated and about possible mandatory vaccination. Vaccine hesitancy and misconceptions about vaccines were also mentioned. Notably, concerns regarding vaccination were more common among participants who reported better health. Personal health concerns and trust in scientists and authorities have previously been identified as important predictors of COVID-19 vaccine acceptance [76]. Thus, the emergence of the theme *Vaccination issues* is not surprising considering the overall good health status and inadequacy of pandemic management indicated by our participants. To increase vaccine acceptance, it therefore appears to be necessary to improve health communication.

Rather alarmingly, we found an increase in the frequency of themes related to deaths and health problems due to COVID-19. The pandemic has brought about a greater confrontation with the theme of death and dying, and this presented a significant burden for some participants. Participants whose family members or friends had died due to COVID-19 reported this event to be particularly stressful. According to the literature, people who have lost a loved one due to COVID-19 represent an at-risk group for the development of mental health problems, as they had to witness a life-threatening disease and were often denied the opportunity to say

their farewell [79, 80]. Our findings underline the importance of continuing to address the challenges and needs of people who have lost a loved one, even in the later phases of the pandemic. Lastly, some of the participants were burdened by their own COVID-19 infection or post-COVID symptoms involving physical but also emotional difficulties. An increased mental health burden has been observed in people with long COVID [81] and further studies are needed to examine this group in greater depth.

Whereas the themes described above were more prominent at T2, others were more frequently mentioned at T1. Of particular note is the decrease in reported financial and work-related burden, which was also evident in the quantitative responses. In the first year of the pandemic, job and income losses, and the consequent impact on mental health, were extensively discussed [27, 82], and largely shaped participants' responses at T1. Over time, the global economic situation has improved but the hardships have remained in economically vulnerable groups [83]. This was also evident in the qualitative part of our study: Although fewer participants indicated work and finances as stressful or negative, those who did reported a significant burden resulting from a co-occurrence of multiple job and work-related stressors such as loans, unemployment, unexpected expenses, etc. Thus, people with substantial losses are likely to remain disproportionately affected throughout the pandemic and in its aftermath, as proposed by the COR theory [35]. Especially in view of recent economic developments (e.g. energy crises), close attention should be paid to economically vulnerable groups to prevent a further deterioration of mental health.

**Positive aspects of the pandemic.** *Reflection and growth* was the most prominent positive theme at both time points. Participants positively evaluated the increased appreciation for life and the opportunity to rethink their priorities. Such new perspectives on life have often been discussed in the context of posttraumatic growth (i.e. positive psychological changes following an adverse life event; [84]). There is some evidence suggesting that perceptions of growth could reflect coping efforts during the COVID-19 pandemic [85]. Although perceived growth might signalise resilience, it might also signalise greater posttraumatic stress symptoms [86, 87], which supports the idea of high variability in individual responses to stressful events. Regarding variability in our study, a lower number of people reported individual or community growth at T2 than at T1 in Austria. The number of Austrian participants recommending cognitive-emotional strategies also decreased, suggesting a tendency towards behaviour-oriented coping in a later phase of the pandemic in Austria. In general, different mentalities and cultural values might also account for different perception of positive aspects across countries.

The themes *Opportunities for meaningful/enjoyable activities* and *Benefits on the interpersonal level* were more common at baseline than at the one-year follow-up. At the beginning of the pandemic, schools were closed and working from home was routine for many. People also travelled less and numerous leisure and work-related activities outside of the home were cancelled. At that time, participants often positively evaluated the increased time resources (e.g. having more time for oneself) and the opportunity to practise hobbies (e.g. reading, gardening). Over time, working and studying from home have become much rarer and almost all activities have gradually been reintroduced into everyday life. Accordingly, at T2, participants likely had less time, and the positive aspects were no longer as pronounced. This assumption is further supported by the higher number of participants who did not report anything positive about the pandemic at T2.

One positive aspect which became more important was digitalisation, with participants appreciating the technological advances during the pandemic and enjoying the opportunity to study and work online. Additionally, some participants at follow-up positively evaluated the possibility of vaccination.

**Population-informed recommendations for dealing with the pandemic.**   A new vaccination-related theme also emerged for the question about recommendations. Predominantly older participants recommended vaccination as a strategy to deal with the pandemic. Given that age and comorbidities are the most important risk factors for severe COVID-19, it is plausible that older people appreciated vaccine availability more than younger ones and perceived it as a solution to end the pandemic, as stated by a woman from Austria: *"Get vaccinated so that we can get the pandemic under control."*

Apart from the vaccination theme, recommendations for dealing with the pandemic did not change much between the two time points. Participants often recommended being physically active, doing enjoyable things, and relaxing regularly. Recommendations to adhere to preventive measures were often combined with comments about trying to live as normally as possible. The latter is likely to promote well-being, as strict adherence to COVID-19 regulations was associated with higher worry and anxiety symptoms [88]. Given the long duration of the pandemic and possible further crises, cultivating personal resources (e.g. by engaging in personally meaningful and pleasant activities) might be essential for maintaining mental health. Additionally, as emphasised by our sample, accepting the situation without losing hope, and trying to find a silver lining, might foster psychological adjustment. Recent literature adds credibility to this idea [89].

## Clinical relevance

Overall, our findings were in line with the concept of psychological flexibility [39] and with the COR theory [35]. Psychological flexibility is an underlying mechanism of the well-established *acceptance and commitment therapy* (ACT; [90]), whereas COR theory promotes the preservation and pursuit of resources to stop loss spirals. Thus, mental health campaigns with elements of ACT and focusing on cultivating resources could boost resilience on a large scale. Such campaigns are strongly recommended given the current global crises and the challenges they pose for the general population.

According to our findings, sources of burden and coping strategies differ across groups (e.g. women vs. men, older vs. younger people, parents vs. non-parents). To provide adequate psychosocial support amidst the pandemic, a close examination of clients' experiences and coping strategies is essential. For example, male participants in our study were more burdened by inadequate pandemic management and communication. Psychoeducation about negative effects of excessive consumption of COVID-related news, and healthy ways to consume media, might increase well-being in this group.

Finally, the observed changes in mental health outcomes, along with the diversity of participants' perceptions and experiences, suggest that psychological adjustment to the pandemic strongly depends on an individual's unique context and resources. Therefore, country-related factors, different challenges in different pandemic phases, as well as personal characteristics and circumstances (e.g. financial resources) need to be equally acknowledged when addressing mental health problems in clinical practice.

## Strengths and limitations

A major strength of this study lies in its longitudinal, mixed-methods design. We used established self-report measures to assess mental health outcomes and combined them with open-ended questions to increase validity. The triangulation of qualitative and quantitative data has allowed us to provide a more nuanced understanding of factors that shape people's perceptions and reactions during the COVID-19 pandemic. Moreover, the study quality was increased by the integration of data on the levels of study design, methods, and interpretation, and the

inclusion of two measurement time points in two different phases of the pandemic. The one-year follow-up enabled us to capture significant changes in participants' pandemic-related experiences, which is not possible in longitudinal studies covering shorter intervals. The diversity of our participants in terms of cultural backgrounds and socioeconomic characteristics further enriched our findings. By comparing five countries, we broadened the knowledge about how and why mental health changes in relation to personal and environmental factors.

However, several limitations need to be acknowledged. First, the generalisability of the findings is limited by the convenience sampling. Future studies using similar methodology in a representative sample could provide important insights. In all countries, the first assessment took place in 2020 and the second in 2021. However, in both waves, some countries (e.g. Croatia) began data collection earlier than others (e.g. Greece), which limits the comparability of the data, especially given the rapidly changing course of the pandemic. Furthermore, due to the lack of pre-pandemic data, it was impossible to assess the causal impact of COVID-19 on mental health. Finally, our findings are limited by the different sample sizes in the countries. The Austrian and Croatian subsamples were considerably larger than the other subsamples, which might have biased the results. Presumably, larger sample sizes would have also allowed for a closer investigation of people with trauma exposure. This remains a subject for future research.

## Conclusion

The present mixed-methods study has enhanced the understanding of psychological responses to the pandemic around Europe. Our findings strongly suggest that adjustment to the pandemic is a dynamic process influenced by an interplay of individual characteristics and circumstances, country-related factors, and the course of the pandemic. For research and clinical practice related to COVID-19 and other crises, we advise adopting a context-sensitive perspective in order to optimally address the mental health needs of the general population and at-risk groups.

## Supporting information

**S1 Appendix. Detailed information on the methodology and participants' characteristics.** (PDF)

**S2 Appendix. Themes and categories (T1 and T2).** (PDF)

**S3 Appendix. Detailed results of the mixed-methods analysis.** (PDF)

## Acknowledgments

The authors thank all team members for their support and contribution to this paper: Hewan Giorgio Ammaturo, Madeleine Jaeger, and Theresa Wagner (team Austria); Jana Kiralj, Ivan Kranjčić, Juraj Sušek, and Andrija Vrhovnik (team Croatia); Niki Adam, Eleftheria Evgeniou, Eirini Fakoureli, Aristoula Maria Kyratzopoulou, Kostas Messas, Triada Palaiokosta, Eleni Papathanasiou, and Konstandina Rapti (team Greece); Sophio Vibliani, and Nana Kharabadze (team Georgia); Luisa Sales, Aida Dias, Guida Manuel, João Veloso, Joana Becker, Diana Andringa, and Rui Brites (team Portugal). We also gratefully acknowledge the work of the German data management team: Laura Kenntemich, Leonie von Huelsen, and Eike Neumann-Runde. Finally, we thank Sarah Mannion de Hernandez for proofreading the paper.

## Author Contributions

**Conceptualization:** Irina Zrnić Novaković, Brigitte Lueger-Schuster.

**Formal analysis:** Irina Zrnić Novaković.

**Investigation:** Irina Zrnić Novaković, Dean Ajduković, Helena Bakić, Camila Borges, Margarida Figueiredo-Braga, Xenia Anastassiou-Hadjicharalambous, Chrysanthi Lioupi, Jana Darejan Javakhishvili, Lela Tsiskarishvili.

**Methodology:** Irina Zrnić Novaković, Dean Ajduković.

**Project administration:** Irina Zrnić Novaković, Annett Lotzin, Brigitte Lueger-Schuster.

**Supervision:** Annett Lotzin, Brigitte Lueger-Schuster.

**Visualization:** Irina Zrnić Novaković.

**Writing – original draft:** Irina Zrnić Novaković.

**Writing – review & editing:** Irina Zrnić Novaković, Dean Ajduković, Helena Bakić, Camila Borges, Margarida Figueiredo-Braga, Annett Lotzin, Xenia Anastassiou-Hadjicharalambous, Chrysanthi Lioupi, Jana Darejan Javakhishvili, Lela Tsiskarishvili, Brigitte Lueger-Schuster.

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
