## [Decision Letter · Decision Letter 0]

10 Feb 2023

PONE-D-22-35029Shaped by the COVID-19 pandemic: Psychological responses from a subjective perspective – A longitudinal mixed-methods study across five European countriesPLOS ONE

Dear Dr. Zrnic Novakovic,

Thank you for submitting your manuscript to PLOS ONE. After careful consideration, we feel that it has merit but does not fully meet PLOS ONE’s publication criteria as it currently stands. Therefore, we invite you to submit a revised version of the manuscript that addresses the points raised during the review process.

We look forward to receiving your revised manuscript.

Kind regards,

Daniel Ahorsu, PhD

Academic Editor

PLOS ONE

Journal Requirements:

When submitting your revision, we need you to address these additional requirements. 1. Please ensure that your manuscript meets PLOS ONE's style requirements, including those for file naming. The PLOS ONE style templates can be found at https://journals.plos.org/plosone/s/file?id=wjVg/PLOSOne_formatting_sample_main_body.pdf and https://journals.plos.org/plosone/s/file?id=ba62/PLOSOne_formatting_sample_title_authors_affiliations.pdf 2. Please ensure that you have specified (1) whether consent was informed and (2) what type you obtained (for instance, written or verbal, and if verbal, how it was documented and witnessed). If your study included minors, state whether you obtained consent from parents or guardians. If the need for consent was waived by the ethics committee, please include this information. 3. We note that the grant information you provided in the ‘Funding Information’ and ‘Financial Disclosure’ sections do not match.  When you resubmit, please ensure that you provide the correct grant numbers for the awards you received for your study in the ‘Funding Information’ section. 4. In your Data Availability statement, you have not specified where the minimal data set underlying the results described in your manuscript can be found. PLOS defines a study's minimal data set as the underlying data used to reach the conclusions drawn in the manuscript and any additional data required to replicate the reported study findings in their entirety. All PLOS journals require that the minimal data set be made fully available. For more information about our data policy, please see http://journals.plos.org/plosone/s/data-availability. Upon re-submitting your revised manuscript, please upload your study’s minimal underlying data set as either Supporting Information files or to a stable, public repository and include the relevant URLs, DOIs, or accession numbers within your revised cover letter. For a list of acceptable repositories, please see http://journals.plos.org/plosone/s/data-availability#loc-recommended-repositories. Any potentially identifying patient information must be fully anonymized. Important: If there are ethical or legal restrictions to sharing your data publicly, please explain these restrictions in detail. Please see our guidelines for more information on what we consider unacceptable restrictions to publicly sharing data: http://journals.plos.org/plosone/s/data-availability#loc-unacceptable-data-access-restrictions. Note that it is not acceptable for the authors to be the sole named individuals responsible for ensuring data access. We will update your Data Availability statement to reflect the information you provide in your cover letter. 5. We note that you have stated that you will provide repository information for your data at acceptance. Should your manuscript be accepted for publication, we will hold it until you provide the relevant accession numbers or DOIs necessary to access your data. If you wish to make changes to your Data Availability statement, please describe these changes in your cover letter and we will update your Data Availability statement to reflect the information you provide. 6. Please include your full ethics statement in the ‘Methods’ section of your manuscript file. In your statement, please include the full name of the IRB or ethics committee who approved or waived your study, as well as whether or not you obtained informed written or verbal consent. If consent was waived for your study, please include this information in your statement as well.  7. Please include captions for your Supporting Information files at the end of your manuscript, and update any in-text citations to match accordingly. Please see our Supporting Information guidelines for more information: http://journals.plos.org/plosone/s/supporting-information.  8. Please review your reference list to ensure that it is complete and correct. If you have cited papers that have been retracted, please include the rationale for doing so in the manuscript text, or remove these references and replace them with relevant current references. Any changes to the reference list should be mentioned in the rebuttal letter that accompanies your revised manuscript. If you need to cite a retracted article, indicate the article’s retracted status in the References list and also include a citation and full reference for the retraction notice.

Reviewers' comments:

Reviewer's Responses to Questions

**Comments to the Author**

1. Is the manuscript technically sound, and do the data support the conclusions?

Reviewer #1: Yes

Reviewer #2: Yes

2. Has the statistical analysis been performed appropriately and rigorously? 

Reviewer #1: Yes

Reviewer #2: Yes

3. Have the authors made all data underlying the findings in their manuscript fully available?

Reviewer #1: Yes

Reviewer #2: Yes

4. Is the manuscript presented in an intelligible fashion and written in standard English?

Reviewer #1: Yes

Reviewer #2: Yes

5. Review Comments to the Author

Reviewer #1: The manuscript contains mixed qualitative and quantitative comparisons of adult mental health in five European countries in a longitudinal study design. The introduction is concise, but exhaustive. The method is well described and allows the replication of these studies. Statistical tests are appropriate for the study hypothesis verification. However, suggest to include effect size for all Student's t-tests, Mann-Whitney's U-tests and Pearson's chi-square tests in the manuscript (see pages 15-16, and Table 3). Figures and tables are helpful, as well as supplementary materials. Discussion and conclusion are well-written and adequate to the data found in the study. Overall, I highly appreciate the content, analyses and hard work to compare five countries in a such large number quantitative and qualitative data.

Reviewer #2: This is a valid study and really helps to understand for clinicians how to navigate therapy and the understandings of the pandemic. I would have loved to see a section of the paper that focusses on post traumatic growth as explaining why not all people were affected negatively and also to show a link between how stressful events can foster resilience.

I did not see a distribution of how many people were involved per country, i think that would be a valuable addition.

6. PLOS authors have the option to publish the peer review history of their article (what does this mean?). If published, this will include your full peer review and any attached files.

Reviewer #1: **Yes: **Aleksandra M. Rogowska

Reviewer #2: No

---

## [Author Response · Author response to Decision Letter 0]

15 Feb 2023

We would like to thank the academic editor and both reviewers for taking the time to review our manuscript. We highly appreciate the positive comments and the valuable suggestions, which have certainly helped us to further improve the quality of the manuscript. 

In the following, please find our responses to the comments raised by the editor and the reviewers:

Response to the academic editor

Response: The paper has been revised to meet PLOS ONE’s style requirements and the file names have been changed accordingly. Moreover, all figures were adapted using the PACE digital diagnostic tool.

 2. Please ensure that you have specified (1) whether consent was informed and (2) what type you obtained (for instance, written or verbal, and if verbal, how it was documented and witnessed). If your study included minors, state whether you obtained consent from parents or guardians. If the need for consent was waived by the ethics committee, please include this information.

Response: We have now specified that written informed consent was obtained from the participants: 

“After receiving information about the study aims, data management, confidentiality, and right to withdraw, all participants provided written informed consent.” (Methods, page 8)

Response: Thank you for pointing out that the information in the ‘Funding Information’ and ‘Financial Disclosure’ sections did not match. As stated in the ‘Financial Disclosure’ in the manuscript, we did not receive any funding for conducting the study, only open access funding will be provided by the University of Vienna, in case the manuscript will be accepted. Thus, the finals statement reads as follows: 

“The authors received no specific funding for conducting the study. Open access funding provided by the University of Vienna.” 

Response: Thank you for reminding us of this important point. The minimal data set underlying the quantitative results is made fully available, along with the experts of the qualitative data relevant to the study. All data can be found in the OSF repository under this link: https://osf.io/p9jx2/; doi: 10.17605/OSF.IO/P9JX2.

Response: As stated earlier, the study’s minimal underlying data set and the supporting information will be deposited to a public repository. The data can be accessed under: https://osf.io/p9jx2/

Response: All details regarding the ethical approval are now provided in the ‘Methods’ section and removed from the supplementary material. The ethics statement in the revised version reads as follows: 

“After receiving information about the study aims, data management, confidentiality, and right to withdraw, all participants provided written informed consent. Ethical approval was obtained in all participating countries: 

Austria: Ethics Committee of the University of Vienna, Reference Number: 00554

Croatia: Ethics Committee of the Department of Psychology, Faculty of Humanities and Social Sciences, University of Zagreb: 21/05/2020

Georgia: Ilia State University, Faculty of Arts and Science, Research Ethics Committee: 12/06/2020

Greece: Social Sciences Ethics Review Board (SSERB), University of Nicosia: SSERB 00109

Portugal: Ethics Committee of the Medical Faculty, University of Porto and Centro Hospitalar São João, Porto, Portugal: CE 201-20”

Response: The in-text citations referring to the supplemental material have been updated. We also added the captions for our Supporting Information at the end of the revised manuscript: 

S1 Appendix. Detailed information on the methodology and participants’ characteristics.

S2 Appendix. Themes and categories (T1 and T2).

S3 Appendix. Detailed results of the mixed-methods analysis. 

Response: The reference list has been reviewed and in-text citations revised. 

Response to the reviewers

Reviewer #1: The manuscript contains mixed qualitative and quantitative comparisons of adult mental health in five European countries in a longitudinal study design. The introduction is concise, but exhaustive. The method is well described and allows the replication of these studies. Statistical tests are appropriate for the study hypothesis verification. However, suggest to include effect size for all Student's t-tests, Mann-Whitney's U-tests and Pearson's chi-square tests in the manuscript (see pages 15-16, and Table 3). Figures and tables are helpful, as well as supplementary materials. Discussion and conclusion are well-written and adequate to the data found in the study. Overall, I highly appreciate the content, analyses and hard work to compare five countries in a such large number quantitative and qualitative data.

Response: We would like to thank the reviewer for the positive evaluation of the manuscript. We highly appreciate it. The suggestion to include effect size is very helpful. In the revised version of the manuscript, the appropriate measures of effect size are provided on pages 15-16 as well as in Table 3. Furthermore, we added a sentence on effect size measures in the ‘Data analysis’ section, which reads as follows: 

“Depending on the analysis, different measures of effect size (Cohen’s d, Odds ratio [OR], Rosenthal’s r, Omega squared [ω2]) were reported.”

Reviewer #2: This is a valid study and really helps to understand for clinicians how to navigate therapy and the understandings of the pandemic. I would have loved to see a section of the paper that focusses on post traumatic growth as explaining why not all people were affected negatively and also to show a link between how stressful events can foster resilience.

I did not see a distribution of how many people were involved per country, i think that would be a valuable addition.

Response: We very much appreciate this positive evaluation of our manuscript and thank the reviewer for the compliments and the helpful suggestions. The number of people per country can be found in Table 1 on page 9, along with relevant sociodemographic characteristics for each of the participating countries. We hope that this table contains all information that you consider necessary. Further details on participants’ characteristics per country are provided in S1 Appendix. The idea to include a section on posttraumatic growth is very valuable and we thank you for proposing it. In the subchapter on positive aspects in the ’Discussion’ section, we now touch upon posttraumatic growth in the following manner: 

“Reflection and growth was the most prominent positive theme at both time points. Participants positively evaluated the increased appreciation for life and the opportunity to rethink their priorities. Such new perspectives on life have often been discussed in the context of posttraumatic growth (i.e. positive psychological changes following an adverse life event; [84]). There is some evidence suggesting that perceptions of growth could reflect coping efforts during the COVID-19 pandemic [85]. Although perceived growth might signalise resilience, it might also signalise greater posttraumatic stress symptoms [86, 87], which supports the idea of high variability in individual responses to stressful events. Regarding variability in our study, a lower number of people reported individual or community growth at T2 than at T1 in Austria.”

---

## [Decision Letter · Decision Letter 1]

16 Apr 2023

Shaped by the COVID-19 pandemic: Psychological responses from a subjective perspective – A longitudinal mixed-methods study across five European countries

PONE-D-22-35029R1

Dear Dr. Zrnic Novakovic,

We’re pleased to inform you that your manuscript has been judged scientifically suitable for publication and will be formally accepted for publication once it meets all outstanding technical requirements.

Kind regards,

Daniel Ahorsu, PhD

Academic Editor

PLOS ONE

Additional Editor Comments (optional):

Congratulations.

Reviewers' comments:

Reviewer's Responses to Questions

**Comments to the Author**

1. If the authors have adequately addressed your comments raised in a previous round of review and you feel that this manuscript is now acceptable for publication, you may indicate that here to bypass the “Comments to the Author” section, enter your conflict of interest statement in the “Confidential to Editor” section, and submit your "Accept" recommendation.

Reviewer #2: All comments have been addressed

2. Is the manuscript technically sound, and do the data support the conclusions?

Reviewer #2: Yes

3. Has the statistical analysis been performed appropriately and rigorously? 

Reviewer #2: Yes

4. Have the authors made all data underlying the findings in their manuscript fully available?

Reviewer #2: (No Response)

5. Is the manuscript presented in an intelligible fashion and written in standard English?

Reviewer #2: Yes

6. Review Comments to the Author

Reviewer #2: (No Response)

7. PLOS authors have the option to publish the peer review history of their article (what does this mean?). If published, this will include your full peer review and any attached files.

Reviewer #2: **Yes: **Seth Mawusi Asafo

---

## [Editor Report · Acceptance letter]

18 Apr 2023

PONE-D-22-35029R1 

Shaped by the COVID-19 pandemic: Psychological responses from a subjective perspective – A longitudinal mixed-methods study across five European countries 

Dear Dr. Zrnić Novaković:

I'm pleased to inform you that your manuscript has been deemed suitable for publication in PLOS ONE. Congratulations! Your manuscript is now with our production department. 

Kind regards, 

on behalf of

Dr. Daniel Ahorsu 

Academic Editor

PLOS ONE